# Multi-UAV Path Planning Based on DRL for Data Collection in UAV-Assisted IoT

Lin Li
*College of Electronic
Information Engineering*
*Inner Mongolia University*
Hohhot, China
linliimu@163.com

Lei Wang
*College of Electronic
Information Engineering*
*Inner Mongolia University*
Hohhot, China
leiwangimu@163.com

Jiawang Hou
*College of Electronic
Information Engineering*
*Inner Mongolia University*
Hohhot, China
jiawanghouimu@163.com

Junjie Ma[*]
*Inner Mongolia Radio Monitoring Station Baotou Branch*
Baotou, China
mjj427@163.com

Yang Liu[*]
*College of Electronic Information Engineering*
*Inner Mongolia University*
Hohhot, China
yangliu@imu.edu.cn

*Abstract*—Due to their flexible mobility and stable network connectivity, unmanned aerial vehicles (UAVs) are increasingly being used as mobile data collectors, greatly expanding the spectrum of data collection. However, safe and effective path planning of multiple UAVs in dynamic environments and complex terrains is always challenging: frequent conflicts arise due to dense flight paths, incomplete observations due to dynamic environments, and risk of local optima from limited exploration. Therefore, we propose a UAV path planning approach based on deep reinforcement learning (DRL). Specifically, we employ the multi-agent proximal policy optimization (MAPPO) algorithm to maximize the data collection rate. We first model the multi-UAV path planning problem as a multi-agent partial observable Markov decision process (MA-POMDP) and integrate the traditional proximal policy optimization (PPO) algorithm into a multi-agent learning framework. Then, to improve the training efficiency of the algorithm and the decision-making capability of the UAVs, the strategy of combining centralized training with decentralized execution is used to enable the effective sharing of information and strategies among UAVs. Furthermore, to mitigate the issue of local optimal convergence during strategy learning due to insufficient exploration of various action plans and strategies in the environment, entropy regularization is introduced into the strategy objective function, enabling the agents to learn more comprehensive and effective path planning strategies. Simulation results validate that the algorithm maximizes total system throughput while adhering to constraints on flight duration, information age, and collision avoidance.

*Index Terms*—Internet of Things, Unmanned aerial vehicle, path planning, deep reinforcement learning, proximal policy optimization

## I. INTRODUCTION

The increasing popularity of wireless devices in the Internet of Things (IoT) has added complexity to the integration with future networks [1], [2]. Unmanned aerial vehicles (UAVs), with their flexible mobility and stable network connectivity, are increasingly being used as mobile data collectors or temporary aerial hubs for IoT endpoints, greatly expanding the spectrum of data collection [3]. With the diversification and increasing complexity of application scenarios, path planning for UAVs has emerged as a critical technological challenge. In dynamic environments and complex terrains, effectively planning paths to avoid obstacles and ensure the successful completion of tasks has become a focal point of current research [4]. Nevertheless, conventional path planning methodologies struggle to accommodate environments characterized by high dynamics and unpredictability [5]. Furthermore, with the increasing number of UAVs, achieving collaborative operations among multiple UAVs to avoid path conflicts remains an urgent issue to be addressed. Therefore, an energy-efficient multi-UAV path planning algorithm is needed to optimize task completion and improve overall system performance.

Objective optimization in the planning of UAV flight paths has been highlighted as an area of interest in recent studies. Traditional node-based algorithms are unable to find workable paths in dynamic environments due to the predetermined graph [6]. In contrast, sampling-based approaches provide more flexibility in the case of dynamic obstacles or environmental changes. The Extended Path Rapidly-Exploring Random Tree (EP-RRT) algorithm was introduced in [7], enhancing the efficiency and convergence of path planning in extended corridor environments using heuristic sampling and the greedy heuristic of RRT-Connect. Although the above algorithms use random sampling of the environment to search for paths, generating feasible paths in narrow passages presents a challenge. To address this issue, [8] introduced a decentralized multi-agent path planning algorithm based on parameterized B-splines. Compared to mathematical-based algorithms, bio-inspired algorithms excel in global search and are better equipped to address the issue of local optima in path planning. A novel particle swarm optimization algorithm was proposed in [9], which addressed the issue of local optima in complex environments through adaptive parameter adjustment and differential evolution operators, enhancing the quality and efficiency of path planning. Bio-inspired algorithms are developed using the structures and behavior found in natural biological systems,

while reinforcement learning algorithms focus on the learning process of UAVs through environmental interactions [10]. A Q-learning-based path planning algorithm was proposed in [11], introducing a shortest distance prioritization policy and a grid-graph-based method, to address the efficient path planning problem for unmanned aerial vehicles (UAVs). Further, [12] introduced a novel reinforcement learning based approach to handle the problem of collision avoidance and optimal trajectory-planning in the context of UAVs in communication networks. Although the aforementioned algorithms effectively address the problem of UAV path planning, demonstrating adaptability to complex environments and flexible decision-making learning, they require a substantial amount of inter-action experience for training, resulting in challenges such as lengthy training times, low sample efficiency, and susceptibility to local optima.

This paper presents a deep reinforcement learning (DRL)-based UAV path planning method. Specifically, we utilize the multi-agent proximal policy optimization (MAPPO) algorithm within an integrated sensing and communication (ISAC)-assisted UAV framework. This innovation effectively addresses challenges faced by existing path planning algorithms, such as the complexity of multi-UAV collaboration, the unpredictability of dynamic environmental changes, the limitations of incomplete observations, and the issue of getting stuck in local optima. To overcome the constraints of UAV detection ranges and environmental observation incompleteness, we model the multi-UAV path planning problem as a multi-agent partially observable Markov decision process (MA-POMDP) and integrate the proximal policy optimization (PPO) algorithm into a multi-agent learning framework for multi-UAV scenarios, accommodating the complex dynamics of multi-UAV operations. Then, to enhance information sharing and strategy coordination among UAVs while increasing their operational flexibility and robustness, we use a strategy that combines centralized training and decentralized execution and utilize the spectrum reuse capability of ISAC to enable UAVs to autonomously select radar and communication modes, further improving the efficiency and decision-making capability of path planning. Moreover, to address the problem of converging to local optima during strategy learning due to insufficient exploration of various action plans and strategies in the environment, entropy regularization is introduced into the strategy objective function, enabling UAVs to learn path planning strategies in a more comprehensive and effective manner.

The remainder of this study is outlined as follows. Section II delineates the system architecture. Section III introduces a deep reinforcement learning (DRL)-based algorithm for integrated communication-sensing scheduling and UAV trajectory optimization. Section IV details the validation of the proposed algorithm via simulation studies. The paper concludes with Section V.

## II. SYSTEM MODEL AND PROBLEM MODELING

In this paper, we explore a system comprised of multiple UAVs equipped with ISAC, operating within a $M \times M \in$

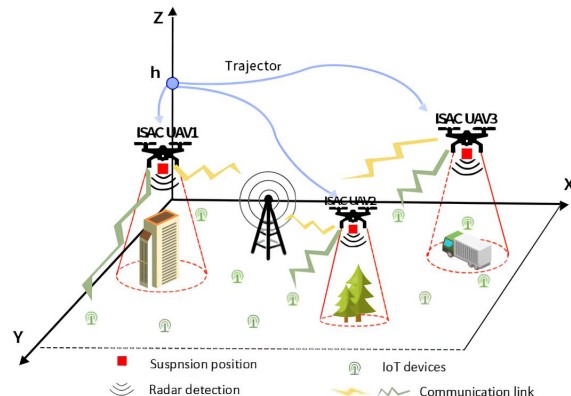

Fig. 1. Multi-UAV-assisted IoT system model.

$N^2$ grid where $N$ belongs to the set of natural numbers. The system consists of $I$ identical dual-function UAVs, and the environment includes specified takeoff and landing locations $\mathcal{L}$ as well as obstacles $\mathcal{B}$. The environment is represented by a tensor $\mathcal{I} \in \mathbb{B}^{M \times M \times 2}$, $\mathbb{B} = \{0, 1\}$, $\mathcal{I} = \mathcal{L} \times \mathcal{B}$. The architecture of this system is depicted in Fig. 1.

### A. UAV Model

UAVs operate at a fixed height $H$, occupying a single cell within this grid. Their movements are carefully regulated to prevent collisions with any obstacles. The movement actions of the $i$-th UAV are $a_{uav,i}(t) \in \tilde{\boldsymbol{a}}_{\boldsymbol{uav}}(u_i(t))$,

$$\tilde{\boldsymbol{a}}_{\boldsymbol{uav}}(u_i(t)) = \begin{cases} \boldsymbol{a}_{\boldsymbol{uav}}, & \text{if } u_i(t) \in \mathcal{L} \\ \boldsymbol{a}_{\boldsymbol{uav}} \backslash a^{land}, & \text{otherwise}, \end{cases} \quad (1)$$

where $\tilde{\boldsymbol{a}}_{\boldsymbol{uav}}(u_i(t))$ is a set of feasible actions that depend on the position of the UAV$_i$. The UAV is limited to moving at a horizontal velocity $V_{uav}$ or standing still, i.e., the velocity of UAV$_i$ at time slot $t$ is $v_i(t) \in \{0, V_{uav}\}$, $t \in [0, T]$, $T \in \mathbb{N}$ is the end time of the UAV's data collection task. The position of the entity changes in accordance with the following motion model:

$$u_i(t+1) = \begin{cases} u_i(t) + a_{\text{uav},i}(t), & \text{if } \phi_i(t) = 1 \\ u_i(t), & \text{otherwise}. \end{cases} \quad (2)$$

The operational state $\phi_i(t) \in \{0, 1\}$ of UAV$_i$, either stationary or in motion:

$$\phi_i(t+1) = \begin{cases} 0, & a_{\text{uav},i}(t) = a^{\text{land}} \lor \phi_i(t) = 0 \\ 1, & \text{otherwise}. \end{cases} \quad (3)$$

The battery energy level $b_i(t) \in \mathbb{N}$ of UAV is defined as,

$$b_i(t+1) = \begin{cases} b_i(t) - 1, & \text{if } \phi_i(t) = 1 \\ b_i(t), & \text{otherwise}. \end{cases} \quad (4)$$

The battery charge of the UAV $b_i(t)$ is initialized at $b(0) \in \mathbb{N}$ and diminishes by one unit with each movement step.

## B. Communication Model

$\text{UAV}_i$ facilitates communication with sensor nodes employing a straightforward time division multiple access (TDMA) method. In this method, during each communication slot $n \in [0, N]$, every sensor node $k \in [1, K]$ selects the data with the highest $\text{SNR}_{i,k}(n)$ for uploading, based on a scheduling algorithm. The TDMA constraint on scheduling variable $q_{i,k}(n) \in \{0, 1\}$

$$\sum_{k=1}^{K} q_{i,k}(n) \leq 1, \quad n \in [0, N], \quad \forall i \in I. \tag{5}$$

The achievable throughput of the $i$-th UAV in a task slot $t$ is the sum of the achievable rates of $K$ sensor nodes during communication slot $n \in [\kappa t, \kappa(t+1) - 1]$

$$C_i(t) = \phi_i(t) \sum_{n=\kappa t}^{\kappa(t+1)-1} \sum_{k=1}^{K} q_{i,k}(n) R_{i,k}(n), \tag{6}$$

where $R_{i,k}(n)$ is the information rate of the $k$-th device at communication slot $n$.

## C. Perception Environment Model

In the UAV flight environment, we consider two environmental features $\boldsymbol{e} = \{e_{\text{weather}}, e_{\text{object}}\}$, where $e_{\text{weather}}$ represents weather conditions and $e_{\text{object}}$ indicates the presence of moving objects near the UAV. For simplicity in data representation, both attributes are quantified on a binary scale, where $\{e_{\text{weather}}, e_{\text{object}}\} \in [0, 1]$. A value of zero is indicative of optimal safety conditions, whereas higher values reflect increasing risks. This study applies a Bernoulli distribution to characterize the probability of an individual UAV encountering high-risk events $X$ at continuous time intervals. This model includes a probability parameter $\alpha \exp(\boldsymbol{e}\boldsymbol{\beta})$, where $\alpha$ is kept fixed and $\boldsymbol{\beta}$ is a vector of the coefficients that should be estimated to forecast risk properly [13].

We also take into account the minimization of the age of information (AoI) of data packets for the UAV to perform communication and perception mode switches quickly based on observed environmental features and also change operational states. Since the maximum size of the packet data queue is $L$, the age of the packets $l$ in the queue is given by $\mathcal{A}_l$, such that together they form the age vector $\boldsymbol{\mathcal{A}} \in \mathbb{R}^L$. Furthermore, the corresponding emergency category of each data packet is $C_l \in \{1, 2, ..., M\}$, such that $\boldsymbol{C} \in \mathbb{R}^L$. The vector $\boldsymbol{\Lambda} \in \mathbb{R}^M$ represents the AoI for each emergency category.

## D. Problem Formulation

The primary goal of the UAV path planning challenge is to optimize data collection efficiency by maximizing throughput, while ensuring compliance with several key constraints: compliance with a prespecified maximum flight duration, obstacle avoidance along the entire flight path and the safe landing within a certain designated area. This optimization problem is defined through the equation outlined below:

$$\max \sum_{t=0}^{T} \sum_{i=1}^{I} \mathcal{C}_i(t) = \max \sum_{t=0}^{T} \sum_{i=1}^{I} [\phi_i(t) \sum_{n=\kappa t}^{\kappa(t+1)-1} \sum_{k=1}^{K} q_{i,k}(n) R_{i,k}(n)], \tag{7}$$

$$\text{s.t.} u_i(t) \neq u_j(t) \vee \phi_j(t) = 0, \forall i, j \in I, i \neq j, \forall t, \tag{7a}$$

$$u_i(t) \notin \boldsymbol{\mathcal{Z}}, \forall i \in I, \forall t, \tag{7b}$$

$$b_i(t) \geq 0, \forall i \in I, \forall t, \tag{7c}$$

$$u_i(0) \in \boldsymbol{\mathcal{L}} \wedge z_i(0) = h, \forall i \in I, \tag{7d}$$

$$\phi_i(0) = 1, \forall i \in I, \tag{7e}$$

$$\sum_{k=1}^{K} q_{i,k}(n) \leq 1, n \in [0, N] \forall i \in I. \tag{7f}$$

(7a) ensures that active UAVs avoid collisions with each other. (7b) mandates that UAVs steer clear of collisions with high structures. (7c) restricts the operating time of the UAV, forcing it to safely land within the landing area before the battery is depleted. (7d) stipulates that the starting position of the UAV is at the takeoff/landing area, with a takeoff height of $h$, while (7e) ensures that the UAV starts in a valid operational state. (7f) is TDMA constraint.

## III. DRL-BASED ALGORITHM

### A. Data Processing

Since the actions of the agents are based solely on their relative positions to features, such as their distance to devices, this paper employs UAV-centric global map mapping processing, which significantly improves learning efficiency. Then the map data can be directly provided to UAVs, and the input space is defined as:

$$\boldsymbol{\Omega} = \underbrace{\mathbb{R}^2}_{\substack{\text{UAV} \\ \text{Position}}} \times \underbrace{\mathbb{B}^{M \times M \times 3}}_{\substack{\text{Environment} \\ \text{Map}}} \times \underbrace{\mathbb{R}^{M \times 2}}_{\substack{\text{Device} \\ \text{Position}}} \times \underbrace{\mathbb{R}^{M \times 2}}_{\substack{Device Data \\ Map}} \times \underbrace{\mathbb{N}}_{\substack{\text{Flying} \\ \text{Time}}}. \tag{8}$$

where $\boldsymbol{p}(t) = [x(t), y(t)]^T \in \mathbb{R}^2$ is the UAV's ground-projected position, $\boldsymbol{\mathcal{I}} \in \mathbb{B}^{M \times M \times 3}$ is the physical environment map in the Boolean field $\mathbb{B} \in \{0, 1\}$, $\boldsymbol{D} \in \mathbb{R}^{K \times 2}$ is the two-dimensional coordinates of $K$ IoT devices, $\boldsymbol{\mathcal{D}} \in \mathbb{R}^{K \times 2}$ is the available data for each device, and $b_t \in \mathbb{N}$ represents the remaining flight time of UAVs. For the purpose of UAV-centric global map processing, the map is enlarged to a dimension of $(2M - 1) \times (2M - 1)$ to ensure UAVs can independently observe the entire map regardless of its position. This expansion centers the map around the UAV's current location, creating the defined input space,

$$\boldsymbol{\Omega}_c = \underbrace{\mathbb{B}^{(2M-1) \times (2M-1) \times 3}}_{\substack{\text{CenteredEnvironment} \\ \text{Map}}} \times \underbrace{\mathbb{R}^{(2M-1) \times 2}}_{\substack{\text{Centered Device} \\ \text{Position}}}$$
$$\times \underbrace{\mathbb{R}^{(2M-1) \times 2}}_{\substack{\text{Centered} Device \\ Data\text{Map}}} \times \underbrace{\mathbb{N}}_{\substack{\text{Flying} \\ \text{Time}}}. \tag{9}$$

### B. Markov Decision Process (MDP)

In an ISAC-assisted multi-UAV framework, we define the combined challenge of communication-sensing scheduling and UAVs path optimization as a partial observable Markov decision process (POMDP). To tackle this, we utilize DRL

techniques that approximate the optimal control strategies for UAVs, despite the lack of initial knowledge about the complex characteristics of wireless channels in densely populated urban settings.

1) State space: To enhance carrier sensing capabilities, the local state information for each agent has been expanded to include $t_{\text{last}}$ and $t_{\text{idle}}$. Here, $t_{\text{last}}$ denotes the number of time steps since the joint radio communication (JRC) system is last utilized to maintain the channel in an idle state, and $t_{\text{idle}}$ indicates the duration in time steps that the channel has remained unoccupied. Another approach to carrier sensing involves the use of channel state $c$, which, in a multi-agent environment, signifies whether a device user is granted priority access to the channel. If a device user successfully transmits data at time step $t - 1$, this user alone is awarded priority ($c = 1$) for the subsequent time step $t$. Each agent is restricted to observing only its immediate state and the local environmental conditions, and the state space available to each agent is structured as follows:

$$\boldsymbol{S}_i = [e_i, a_i, u_i, \boldsymbol{\Lambda}_i, t_{\text{last},i}, t_{\text{idle}}, \mathbf{u}_i, \boldsymbol{\mathcal{I}}, \boldsymbol{D}, \boldsymbol{\mathcal{D}}, b_i]. \qquad (10)$$

The system's state space is composed by aggregating the observation spaces of each of the $I$ agents:

$$\boldsymbol{S} = [e_1, a_1, u_1, \boldsymbol{\Lambda}_1, t_{\text{last},1}, \mathbf{u}_i, ..., e_I, a_I, u_I, \boldsymbol{\Lambda}_I, t_{\text{last},I}, \mathbf{u}_i, t_{\text{idle}}, \boldsymbol{\mathcal{I}}, \boldsymbol{D}, \boldsymbol{\mathcal{D}}, b]. \qquad (11)$$

2) UAV action space: Agents are allowed to cooperate by foregoing their access to the communication channel, thus broadening the range of possible actions to include a non-operational action,

$$\boldsymbol{A}_i = \{\text{nop}, \boldsymbol{a}_{uav}, a^{\text{r}}, a^{(1)}, a^{(2)}, ..., a^{(M)}\}, \qquad (12)$$

where the subscript $i$ is the index of the given agent.

3) Reward: At each time step $i$, the reward structure for the agent is designed as a weighted sum that promotes efforts to minimize the age of data packets in the queue while also aiming to enhance throughput.

$$\begin{aligned} r_i(t) = {} & w_{i,\text{age}} r_{i,\text{age}}(t) + w_{i,\text{over}} r_{i,\text{over}}(t) + w_{i,\text{rad}} r_{i,\text{rad}}(t) \\ & + w_{i,\text{data}} r_{i,\text{data}}(t) + w_{i,\text{sc}} r_{i,\text{sc}}(t) + w_{i,\text{mov}} r_{i,\text{mov}}(t) \\ & + w_{i,\text{crash}} r_{i,\text{crash}}(t), \end{aligned} \qquad (13)$$

where $w_{i,\text{age}}$, $w_{i,\text{over}}$, $w_{i,\text{rad}}$, $w_{i,\text{data}}$, $w_{i,\text{sc}}$, $w_{i,\text{mov}}$ and $w_{i,\text{crash}}$ are weights. $r_{i,\text{age}}$ encourages agents to minimize AoI as much as possible, encourages agents to send packets before queue overflow occurs, $r_{i,\text{rad}}$ encourages agents to perform radar scans when environmental conditions are unfavorable, $r_{i,\text{data}}$ is the reward for data collection based on the throughput achieved during the current period, $r_{i,\text{mov}}$ is the penalty for UAVs continuing to move without completing tasks, and $r_{i,\text{crash}}$ is the penalty for UAVs failing to safely land in the landing zone when the remaining flight time is zero.

Upon evaluating the data transmission and expiry within the current time step, the penalty for exceeding the queue's capacity is calculated based on the number of new data packets that cannot be accommodated. This penalty scales in

**Algorithm 1** MAPPO-based algorithm for ISAC-assisted UAVs path planning

---
1: **for** episode = 1 to Num episodes **do**
2:     Run the agents in the environment according to policy $\pi_{\theta_{\text{old}}}$
3:     Calculate the advantage estimate $A$ for each agent using equation (17)
4:     **for** step = 1 to $T$ **do**
5:         **for** minibatch with size $M < N/K$ **do**
6:             Sample mini-batch of experiences (states, actions, rewards)
7:             Compute advantage estimates $A_i$ and discounted returns $R_i$
8:             Compute policy probabilities $\pi_{\theta_{\text{new}}}(a_i \mid s_i)$
9:             Compute surrogate objective function $\mathcal{L}^{CLIP}(\theta) + r(\tau)$ as per equation (19)
10:            Update policy parameters $\theta$ by optimizing $\mathcal{L}^{CLIP}(\theta) + r(\tau)$
11:         **end for**
12:     **end for**
13: **end for**

---

direct proportion to the excess packets, reflecting the system's inability to manage the overflow effectively,

$$\begin{aligned} r_{i,over}(t) = \min(0, (\sum_{l=1}^{L} [\mathrm{I}_{\mathbb{R}+}(u_l(t)) - \mathrm{I}_{\mathbb{R}+}(a_l(t) - \mathcal{A}_{\max}) - T_l(t)] \\ + \sum_{m=1}^{M} \Upsilon^{(m)}(t) - N)). \end{aligned}$$

$$(14)$$

Whenever a high-risk event $X$ occurs, the number of adverse environmental characteristics in $r_{i,rad}(t)$ and $e$ is proportional:

$$r_{i,rad}(t) = -(e_{\text{weather}}, e_{\text{object}}) \times X(t). \qquad (15)$$

*C. Algorithm Design Based on MAPPO*

In multi-agent scenarios characterized by complexity, traditional Q-learning applied individually to agents typically results in inadequate performance [14]. To address the challenges posed by partially observable Markov decision processes, we propose a MAPPO algorithm. This algorithm leverages the strengths of policy gradient techniques, which are notably advantageous because they do not require knowledge of the state transition probabilities $P(s_{t+1} \mid s_t, a_t)$. This freedom from the Markov assumptions of the preceding states is a significant methodological advantage. Although as a rule agent-based systems utilize inter-agent communication or centralized frameworks, our method introduces a decentralized multi-agent PPO algorithm. This innovation is aimed at improving the effectiveness without the necessity to have a centralized control or direct interaction among the agents.

In a multi-agent setting with $I$ agents, we describe $\boldsymbol{\pi} = \{\pi_1, \pi_2, \ldots, \pi_I\}$ as the set of policies, parameterized by $\Theta = \{\Theta_1, \Theta_2, \ldots, \Theta_I\}$. For each iteration of the algorithm,

the policy of each agent can be trained independently. To improve the knowledge sharing among the agents, we suggest the construction of sequence of policy models, represented by $I^-$ with $I^- < I$. Each policy $\pi_{i-}$ is assigned to a subset of agents in the environment. Therefore, the objective function of the policy function is as follows:

$$L^{CLIP}(\Theta_{i-}) = E_{o_i, a \sim \pi_{i-}}[\min(\delta(\Theta_{i-})\hat{A}_i^{\pi_{i-}}(t),$$
$$\max(1 - \epsilon, \min(\delta_t(\Theta_{i-}), 1 + \epsilon))\hat{A}_i^{\pi_{i-}}(t))], \quad (16)$$

where $\delta_t(\Theta_{i-}) = \frac{\pi_{i-}(a_t|o_t)}{\pi_{i-,old}(a_t|o_t)}$ is the ratio of the policy $\pi_{i-}$ change between the previous and current iterations. $\hat{A}_i^{\pi_{i-}}(t)$ represents the estimated advantage value:

$$\hat{A}_i^{\pi_{i-}}(t) = r_i(t) + \gamma \hat{V}_i^{\pi_{i-}}(s(t+1)) - \hat{V}_i^{\pi_{i-}}(s(t)). \quad (17)$$

The objective function of the parameterized value function $\Theta_{i-}$ is:

$$L^{VAL}(\Theta_{i-}) = E_{o_i, a \sim \pi_{i-}}[(\hat{V}_i^{\pi_{i-}}(o_i(t)) - y_i(t))^2], \quad (18)$$

where $y_i(t) = r_i(t) + \gamma \hat{V}_i^{\pi_{i-}}(s(t+1))$ is the target value. While this framework necessitates the sharing of policy parameters among the agents during the training phase, it supports autonomous decision-making by each agent when operational.

In policy-gradient learning frameworks, the exploration of new policies is performed by sampling actions from the recent modifications of the stochastic policy. The smell of diminishing randomness of policy selection in the training stage usually results in poor exploration that confines the system to local optimal solutions. To overcome this limitation and ensure a more comprehensive exploration, this paper introduces an entropy regularization component into the objective function. This modification increases the exploration power of the agents that prevents the local optima and enhances the overall system performance,

$$L^{CLIP+VAL}(\Theta) = E_t[L^{CLIP}(\Theta) + k_1 L^{VAL}(\Theta)$$
$$+ k_2 S[\pi_{\Theta_{i-}}](o_i)], \quad (19)$$

where $k_1$ and $k_2$ are coefficients, and $S[\pi_{\Theta_{i-}}](o_i)$ represents the policy entropy of $\pi_{\Theta_{i-}}$ given input observation $o_i$. The MAPPO-based algorithm for ISAC-assisted UAVs path planning is illustrated in Algorithm 1.

## IV. SIMULATION ANALYSIS

TABLE I. Simulation parameters of the system

| Parameter Name | Parameter Value |
| --- | --- |
| Grid Area Size | $320 \times 320$ |
| Grid Cell Size | $10m \times 10m$ |
| UAV Flight Speed $V_{uav}$ | $2.5m/s$ |
| $\gamma_{los}$, $\gamma_{Nlos}$ | 2.27, 3.64 |
| $\sigma_{los}^2$, $\sigma_{Nlos}^2$ | 2, 5 |
| Data Queue Length $L$ | 3 |
| Data Packet Threshold Age $\mathcal{A}_{max}$ | 2 |

TABLE II. Network hyperparameters of MAPPO

| Parameter Name | Parameter Value |
| --- | --- |
| Number of Convolutional Layers | 2 |
| Number of Hidden Layers | 3 |
| Learning Rate $l_r$ | 0.0001 |
| Discount Factor $\gamma$ | 0.95 |
| Batch Size | 32 |
| Activation Function | ReLU |

In this experiment, the binary exponential backoff (BEB) algorithm, dueling double deep Q-network (D3QN) algorithm, and advantage actor-critic (A2C) algorithm are used as comparative algorithms. The system parameters are set as shown in Table 1. The experimental setup involves UAVs tasked with data collection within a 320×320 grid, initiated with a total available flight duration of $T$ steps. For each UAV action, either movement or hovering, there is a decrement of one step in the remaining flight time. Regulatory compliance mandates that the UAV cannot operate over tall buildings or beyond the confines of the designated grid. Each operational slot $t \in [0, T]$ encompasses $\kappa = 4$ communication slots $n \in [0, N]$. To enhance data freshness, the system's threshold for maximum

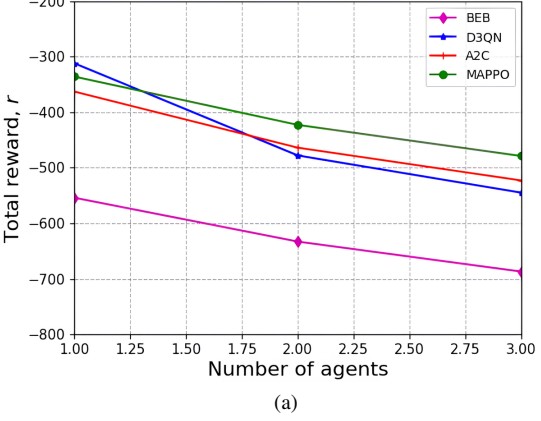
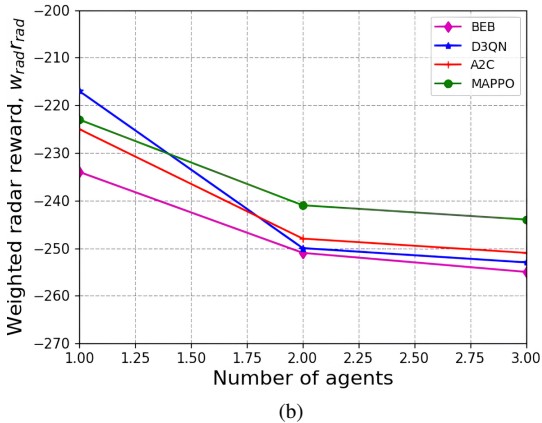

Fig. 2. Effect of the number of UAVs on (a) total reward (b) radar reward.

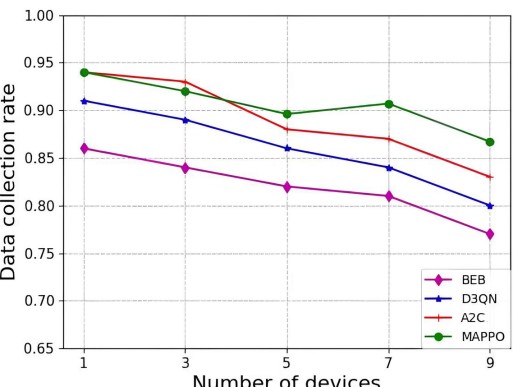

Fig. 3. The effect of the number of IoT devices on data collection rate.

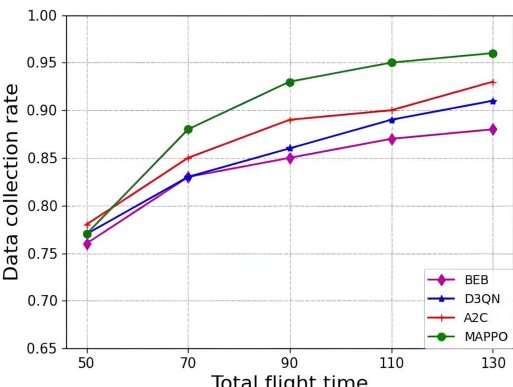

Fig. 5. The effect of the total flight time on the data collection rate.

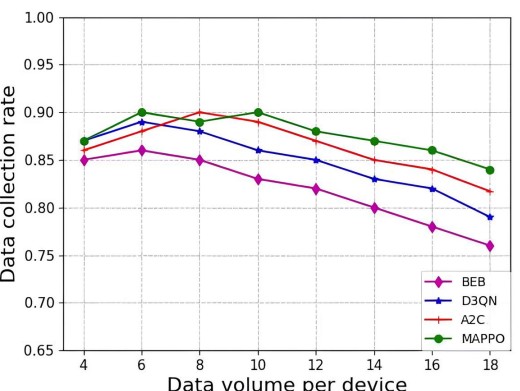

Fig. 4. The effect of the data volume per device on the data collection rate.

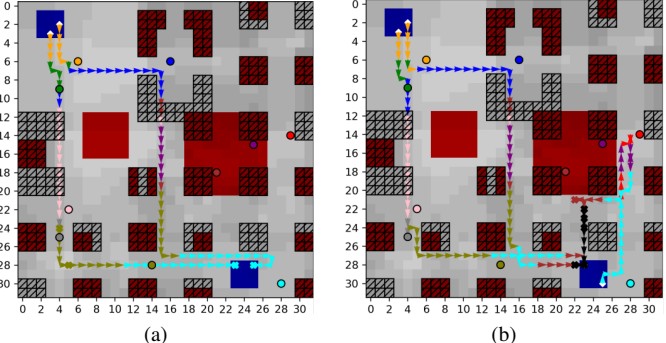

Fig. 6. Influence of the number of UAVs on the flight path (a) $I = 2$ (b) $I = 3$.

age has been lowered to $\mathcal{A}_{\max} = 2$, preventing the aging of data packets in the queue.

The hyperparameter settings for the MAPPO network are shown in Table 2. The training regimen is structured around 400 time steps per episode, with a total of 2500 episodes conducted. To validate the stability and reproducibility of the experimental outcomes, each parameter configuration is rigorously tested using multiple independent trials, each initiated with a unique random seed.

Fig. 2 demonstrates the impact of the number of UAVs on the performance of different algorithms. We can observe that the performance of each algorithm decreases as the number of UAVs increases. First, as the number of UAVs increases, the frequency of radar mode switching increases accordingly, which raises the risk of being penalized. As shown in Fig. 2(b), this leads to a decrease in radar-related rewards. Additionally, more UAVs in the system increase the competition for a limited time step to transmit data or operate the radar. And the agents frequently switch radar modes, this leads to a decrease in communication performance and ultimately to a decrease

in the overall reward, as shown in Fig. 2(a). However, as the increasing number of UAVs, the MAPPO method maintains optimal performance and better balances the increase in radar operations with the need for normal transmission of data streams. Therefore, the MAPPO approach enables agents to learn effective action strategies for the current environmental conditions with minimal prior knowledge of the environment.

Fig. 3 analyzes the impact of the number of IoT devices on the average data collection rate for successful UAV landings. Due to the random placement of IoT devices in unoccupied areas of the map, an increase in the number of devices results in more complex trajectory requirements and performance degradation. From Fig. 3, it is observed that when devices are less than 5, the proposed algorithm does not achieve the optimal data collection rate. However, when devices reach or exceed 5, the performance of the proposed algorithm remains optimal. This indicates that the proposed algorithm has good adaptability to an increasing number of devices.

Fig. 4 shows the effect of the initial amount of data per device on the overall data collection rate. We find that as the number of UAVs increases, the data collection rate increases for all algorithms. However, beyond 10-12 data units, increasing the initial amount of data per device leads to a

decrease in the data collection rate. This is because the UAVs are limited by a maximum flight time and are prioritized to land before exceeding this limit, thus forgoing some data collection. In addition, we note that the data collection rate of the proposed algorithm decreases at a slower rate as the initial amount of data per device increases. This indicates that the proposed algorithm can collect more device data while satisfying the need to land before the end of the maximum flight time. This is attributed to the fact that the algorithm employs entropy regularization, which enhances exploration and allows for a more effective action selection strategy in response to an increase in the amount of initial data per device.

Fig. 5 illustrates the relationship between the maximum allowable flight time and the data collection rate. It is evident that an increase in the maximum flight time correlates with an improvement in the rate of data collection. Among the four different algorithms, the data collection rate of the MAPPO algorithm is the highest. This is because the MAPPO algorithm can more effectively utilize each data sample, meaning that within the same timeframe, MAPPO can learn more environmental information, thus collecting more data. However, as the bulk of the data is secured, this effect diminishes, leading UAVs to focus more on minimizing total flight time and ensuring a safe landing rather than gathering the residual data.

Fig. 3, 4, and 5 illustrate that as scenario parameters increase, particularly at higher values, the MAPPO algorithm we propose consistently achieves a higher data collection rate compared to other algorithms. The BEB algorithm's random action selection fails to adopt effective strategies in response to environmental parameter changes. Additionally, an expansion of the UAV's action space with increasing scenario parameters leads to a decline in the performance of value-based D3QN algorithms. Although A2C is a policy gradient method, its high variance and sensitivity to non-linear gradients limit its effectiveness. Our proposed MAPPO algorithm constrains the magnitude of policy updates by estimating the policy gradient and incorporates an entropy regularization term into the objective function to enhance the agent's exploration capabilities. Consequently, the algorithm enables the agent to adapt well to changes in scenario parameters, thereby making effective action selection strategies in response to current parameter variations.

The range of values for random scenario parameters is as follows: the number of deployed UAVs $I \in \{2, 3\}$., the number of IoT sensors $K \in [310]$, the amount of data to be collected by each device $\mathcal{D}_k \in [5.020.0]$, and the maximum flight time steps. The positions of IoT devices are randomly distributed throughout the entire unoccupied map space. In Fig. 6, other scenario parameters remain constant, and two scenario instances, labeled 6(a), and 6(b), were selected to illustrate how path planning adapts to the increase in the number of UAVs. In Fig. 6(a), two UAVs are deployed. Due to the maximum flight time limit, they ignore the red, purple, and brown devices, collect data from other devices, and land in the bottom-right landing area, achieving a collection rate of 77.3%. In Fig. 6(b), three UAVs are deployed, two from the top-left and one from the bottom-right, collecting data from all devices with a collection rate of 100%. The three UAVs evenly distribute data collection tasks, achieve comprehensive data collection, avoid tall buildings, and land along efficient trajectories promptly.

## V. Conclusion

In this paper, in order to address the challenges of limited UAV detection range and incomplete environmental observations, we model the multi-UAV path planning problem as an MA-POMDP and integrate the PPO algorithm into a multi-agent learning framework, proposing a MAPPO-based algorithm to enhance data collection rates. To improve the training efficiency of the algorithm and the decision-making capabilities of the UAVs, the strategy of combining centralized training with decentralized execution is used to enable the effective sharing of information and strategies among UAVs. Additionally, to mitigate the problem of converging on local optima due to insufficient exploration during strategy learning, entropy regularization is introduced to the strategy objective function. Simulation results show that compared to competitive algorithms, the proposed algorithm can significantly improve the throughput of the system, thereby enhancing data collection rates.

This study is designed to maximize system throughput, and while UAV energy consumption is considered under constraints, it has not been quantitatively analyzed. Considering the limited battery life and payload capacity of UAVs, it is crucial to tackle the issue of reducing overall energy consumption in UAV-assisted IoT systems to extend their operational lifespan. Therefore, in future research, multi-objective optimization schemes for throughput and energy efficiency need to be designed to ensure that the system is better adapted to practical applications.

## Acknowledgment

The authors are grateful to the National Science Foundation of China for its support of this research. This work is supported by the National Natural Science Foundation of China under grants 62161037 and 62071257, and is supported in part by the Natural Science Foundation of Inner Mongolia Autonomous Region under Grants 2023JQ17.

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
