# OpenReview forum: "Multi-UAV Path Planning Based on DRL for Data Collection in UAV-Assisted IoT"
_IEEE.org/ICIST/2024/Conference — IEEE ICIST 2024 Conference Submission_

### Official Review · Reviewer_LAfK · 2024-08-22
**This article is quite fascinating and of high quality.**

**Rating:** 7
**Confidence:** 3

**Review:**

This article, " Multi-UAV Path Planning Based on DRL for Data Collection in UAV-Assisted IoT," proposes a UAV path planning method based on deep reinforcement learning (DRL). Firstly, the multi-agent near-end Strategy Optimization (MAPPO) algorithm is used to ensure effective path planning for UAVs. Finally, a strategy combining centralized training and decentralized execution is adopted to enable UAVs to share information and strategies for path planning collaboration. The article has clear logic and organization, but there are still some problems. My specific feedback is as follows :1) In the abstract part, the author should briefly discuss the problem in the first two sentences. 2) In the introduction, the author has insufficient background content for the study of UAV path planning. 3) What are the advantages of combining multiple algorithms in the research?

---

### Official Review · Reviewer_pUXV · 2024-08-22
**This article is well written and acceptable.**

**Rating:** 7
**Confidence:** 4

**Review:**

1.Authors are required to review manuscript templates and make corrections.
2.There are many grammatical and typographical errors in the manuscript. Please check the full text carefully and correct them.
3.The formula in the manuscript is too small, please change it.
4.Based on the proposed idea and obtained results in this paper, the authors should be able to present some more descriptions in conclusion part, for example, further study direction.

---

### Official Review · Reviewer_5vcb · 2024-08-24
**This paper proposed a UAV path planning approach based on deep reinforcement learning (DRL). Specifically, adopting the multi-agent proximal policy optimization (MAPPO) algorithm to ensure efficient UAV path planning, thereby maximizing the data collection rate. The topic of this paper is interesting. Below is a list of comments that should be taken into account further when revising the paper.**

**Rating:** 7
**Confidence:** 3

**Review:**

1. The contribution of this article should be compared with previous literature, and the basic technical difficulties of this article should be listed? And what methods should be used to solve this problem, emphasizing novelty and technological contribution.
2. In the experimental results section, after separately explaining the four different algorithms, they should be compared uniformly to draw an overall conclusion.
3. In the conclusion section, all strategies should be compared. Finally, a MAPPO-based algorithm is the most effective feature extraction component to improve the model’s expressive ability to make the best model performance. Meanwhile, please elaborate on the future plans.

---

### Decision · Program_Chairs · 2024-09-06

Accept (Oral)